Are we throwing away good data? Evaluation of chimera detection algorithms on long-read amplicons reveals high false-positive rates across algorithms

Hakimzadeh Ali ali.hakimzadeh@ut.ee 1
Mikryukov Vladimir 1 2
Metsoja Martin 1
Tedersoo Leho 1 2 3
Anslan Sten sten.s.anslan@jyu.fi sten.anslan@gmail.com 2 4 5
1 Institute of Ecology and Earth Sciences, University of Tartu , Tartu , Estonia
2 Mycology and Microbiology Center, University of Tartu , Tartu , Estonia
3 Department of Zoology, College of Science, King Saud University , Riyadh , Saudi Arabia
4 Department of Biology, College of Science, Princess Nourah bint Abdulrahman University , Riyadh , Saudi Arabia
5 Department of Biological and Environmental Science, University of Jyväskylä , Jyväskylä , Finland
Nunes-da-Fonseca Rodrigo
Electronic publication date: 2025 Dec 5
Publication date: 2025
Volume: 13
Electronic Location ID: e20456
Received 2025 Jul 9; Accepted 2025 Oct 31
Copyright: ©2025 Hakimzadeh et al.
Copyright year: 2025
Copyright holder: Hakimzadeh et al.
License: This is an open access article distributed under the terms of the Creative Commons Attribution License, which permits unrestricted use, distribution, reproduction and adaptation in any medium and for any purpose provided that it is properly attributed. For attribution, the original author(s), title, publication source (PeerJ) and either DOI or URL of the article must be cited.
License URL: https://creativecommons.org/licenses/by/4.0/

Keywords: Long-read sequencing, De novo chimera detection, Metabarcoding, Bioinformatics, rRNA, Internal transcribed spacer

Funding: The Estonian Science Foundation MOBERC116 The Research Council of Finland Decision number 362828 Researchers Supporting Project at Princess Nourah bint Abdulrahman University (Riyadh, Saudi Arabia) HCPNU2025R402 This work was supported by the Estonian Science Foundation (grant MOBERC116), the Research Council of Finland (Decision number 362828) and Researchers Supporting Project (HCPNU2025R402) at Princess Nourah bint Abdulrahman University (Riyadh, Saudi Arabia). The funders had no role in study design, data collection and analysis, decision to publish, or preparation of the manuscript.

==============================
Long-read amplicon sequencing has enabled us to return to full-length DNA barcodes, which benefit from the higher taxonomic resolution in metabarcoding-based biodiversity studies. However, chimeric sequences (artificial constructs formed when incomplete amplicons fuse during polymerase chain reaction (PCR)) remain challenging, potentially skewing diversity estimates and ecological inferences. Here, we benchmark three de novo chimera detection algorithms, uchime_denovo, removeBimeraDenovo, and chimeras_denovo, on simulated and empirical eukaryotic full-ITS (rRNA ITS1-5.8S-ITS2) datasets to evaluate their precision, sensitivity, and effects on the final OTUs composition/community structure. Upon simulated data, uchime_denovo achieved the highest precision even with default settings, whereas other algorithms displayed high false-positive chimera rates without setting adjustments. Similarly, the tests upon empirical data showed that uchime_denovo had lower false positive rates, whereas about half of the sequences in the putative chimeric batch were false positives when using chimeras_denovo and removeBimeraDenovo. We found that most of the false-negative chimeras contained multiple 5.8S regions, indicating PacBio library preparation artifacts rather than PCR artifacts. However, OTU-level comparisons indicated that overall richness and community-ordination patterns remain largely consistent across different chimera-filtering approaches with or without accounting for false positives and negatives.

Introduction

Metabarcoding is a molecular technique that uses high-throughput sequencing (HTS) techniques to simultaneously identify numerous species in multiple environmental samples (Taberlet et al., 2012). By targeting specific gene fragments (markers, barcoding regions), metabarcoding has become a widespread tool in biodiversity research. It has offered new insights into biogeography patterns of, e.g., fungi, bacteria, and animals (Tedersoo et al., 2014; Bahram et al., 2018; Mathon et al., 2022; Abrego et al., 2024) and species diversity, including diversity of ‘dark taxa’, across various ecosystems (Boussarie et al., 2018; Tedersoo et al., 2020).

Metabarcoding workflows rely on amplifying and sequencing barcoding gene fragments to identify taxa within corresponding DNA samples. The most common approach is the second-generation, short-read (up to 550-bp amplicons) metabarcode sequencing workflow via the Illumina platform. The more recently developed third-generation, long-read platforms, PacBio (Rhoads & Au, 2015) and Oxford Nanopore (Jain et al., 2016), are becoming increasingly popular because of increasing throughput and read quality improvements. Amplifying and sequencing full-length barcodes (long-reads) provides finer resolution in delineating species and assigning higher taxonomic levels with greater confidence (Tedersoo et al., 2021).

One of the most common sequencing artefacts is the formation of chimeric DNA fragments through incomplete template extension or template switching, where partially extended sequences anneal to other templates during amplification (Kebschull & Zador, 2015; Odelberg et al., 1995; Von Wintzingerode, Göbel & Stackebrandt, 1997). Compared with short reads, longer sequences are more prone to chimera formation because long amplicons are more likely to be interrupted; they bear more potential chimera breakpoints and may require additional polymerase chain reaction (PCR) cycles, which all increase the risk of chimerism (Heeger et al., 2018; Tedersoo, Tooming-Klunderud & Anslan, 2018). If not filtered out during bioinformatic sequence data processing, then reads from those artificial molecules lead to inaccuracies by skewing diversity estimates in the dataset (Aas, Davey & Kauserud, 2017; Nilsson et al., 2010).

Several software suites, such as USEARCH (Edgar, 2010), VSEARCH (Rognes et al., 2016), and DADA2 (Callahan et al., 2016), that have been developed to process metabarcoding (amplicon) data, additionally host algorithms that address filtering of putative chimeric sequences. Those chimera filtering algorithms are currently used in most bioinformatic software for metabarcoding data analyses (Hakimzadeh et al., 2023). The most utilized one, UCHIME (Edgar et al., 2011), has demonstrated high sensitivity in chimera filtering with short sequences (∼250 bp). However, with the rapid developments in third-generation sequencing methods, the metabarcoding workflows are no longer restricted to minibarcodes (i.e., short reads). Although the settings of UCHIME can be adjusted to optimize the sensitivity and specificity, the default options benchmarked for short-reads are often applied for long-reads (Fichot & Norman, 2013; Mosher et al., 2013). The same applies to a chimera removal algorithm (removeBimeraDenovo) within the DADA2 software suite (Furneaux et al., 2021). A new (experimental) algorithm, chimeras_denovo (Rognes, 2023), has been integrated into VSEARCH v.2.23.0 (July 2023). This algorithm was designed explicitly to filter chimeric reads from long-read sequencing sets. The increasing application of long-read sequencing technologies in metabarcoding studies necessitates the evaluation of chimera filtering algorithms to evaluate their performance on longer reads than those benchmarked initially.

Herein, we evaluate the effectiveness of uchime_denovo, chimeras_denovo (both, as implemented in VSEARCH v2.29.4), and removeBimeraDenovo (DADA2 v.1.32) algorithms on long-read, PacBio, metabarcoding datasets. This assessment utilizes simulated and empirical full-length rRNA ITS (internal transcribed spacer) datasets.

Material and Methods

Simulated data

To test the chimera detection approaches on long reads, we first constructed a simulated PacBio dataset based on the reference 186 full-ITS (ITS1-5.8S-ITS2) sequences obtained from the EUKARYOME v.1.9.2 database Tedersooetal2024 related to the study of Jamy et al. (2022). The reference sequences, annotated at the species level, are listed in Table S1. The simulated dataset was generated with SimLoRD v1.0.4 (Stöcker, Köster & Rahmann, 2016), a package designed for simulating PacBio data with characteristic errors. For each of the 186 ITS sequences, we assigned a unique read count using the -n parameter in SimLoRD, calculated to achieve proportionate representation according to sequence properties per input. This included the creation of 1,000 to 9,000 full-ITS sequences per input sequence while maintaining sequence length variation typical of the original data. SimLoRD was executed with up to 30 passes to create enough variability and simulate sequencing complexity similar to a real-world setting (Table S2). SimLoRD outputs FASTQ-format reads with simulated per-base quality scores. The total pool of simulated dataset (424,010 sequences) was then subjected to quality filtering (VSEARCH --fastq_filter --fastq_qmax 93, --fastq_maxee 1), and followingly, ITSx v.1.1.3 (Bengtsson-Palme et al., 2013) was used to clip flanking primer binding sites by keeping only full-length ITS (--nhmmer TRUE, -E 1e-2, --complement TRUE, --only_full TRUE). These preprocessing steps eliminated low-quality, truncated reads and reduced the dataset to 44,470 full-length ITS sequences.

The latter dataset was supplemented with in-silico chimeras. The chimeric reads were generated using a custom Python script (Table S2) that creates chimeras by randomly selecting pairs of sequences, determining breakpoints based on ITS region-specific probabilities, and applying length constraints to ensure realistic sequence properties. In addition, it ensures that the chimeras do not dominate in abundance and length compared to the original (parent) sequences. A total of 2,484 chimeras were generated (Table S3). After adding chimeras to the simulated dataset, the total number of sequences was 46,954 (28,989 unique sequences and 6.2% chimeric reads). This approach provided a dataset where the origin of each sequence, whether parental or chimeric, is known.

Scoring chimera detection

To examine the precision and recall of uchime_denovo and chimeras_denovo, a series of runs with different parameter settings were executed with our simulated dataset. Here, sequences flagged as ‘borderline’ by uchime_denovo were considered chimeras. The removeBimeraDenovo function (of the DADA2 package) was tested only with default settings. Precision is the number of detected true chimeras (true positives) divided by the number of all chimeras returned by the chimera filtering process. This can be expressed as true positives/(true positives + false positives). Recall is the proportion of true positives detected correctly over the total number of values (number of sequences) classified as true positives and false negatives (true positives/true positives + false negatives). We used the F1 score, a measure that evaluates the performance of an algorithm by considering both precision and recall. F1=2×precision×recallprecision+recall

Various tested adjustable parameter settings are listed in Table 1. We tested a combination of 49 runs for uchime_denovo and 22 runs for chimeras_denovo.

Table 1 Adjusted uchime_denovo and chimeras_denovo parameters to examine the precision and recall of the chimera detections on the simulated dataset.

	Parameter	Default value	Tested ranges	Description	
uchime_denovo	--dn	1.4	1.4–2.0 (step 0.2)	Pseudo-count cutoff for nucleotide difference before assigning as chimeric. A higher value increases specificity, reducing false positives.	
--mindiffs	3	2–4 (step 1)	Minimum differences required per query segment to consider a chimera. Higher values reduce the likelihood of detecting chimeras.	
--minh	0.28	0.10–0.28 (step 0.02), 0.10–0.05 (step 0.01)	Threshold for considering a sequence as chimera. Smaller values improve sensitivity.	
--abskew	2	2–16 (step 1)	Minimum abundance ratio between the parent sequence and chimera. Reflects abundance skew in the dataset.	
--xn	2	2, 3	Controls how strongly the algorithm weighs instances where a sequence is not classified as a chimera.	
--mindiv	0.8	0.4, 0.6	The divergence threshold below which sequences are not considered chimeric. Lower values increase sensitivity.	
chimeras_denovo	--chimeras_diff_pct	0	0.5–0.9	Percentage mismatch allowed in chimeric regions. Larger values increase sensitivity to sequence mismatches.	
--chimeras_length_min	10	10–60 (step 10)	Minimum length of chimeric regions. Longer regions reduce false positives.	
--chimeras_parts	1	2, 3	The number of parts a sequence is divided into and adjusted to test segmentation effects on detection.	
--abskew	1	2–6 (step 1)	Minimum abundance ratio	

Empirical dataset

We used a dataset from Jamy et al. (2022) (BioProject https://www.ebi.ac.uk/ena/browser/view/PRJEB45931) for testing chimera filtering algorithms on an empirical ITS amplicons dataset. The samples obtained in this investigation were derived from various environments, encompassing nine from marine (marine water), four from freshwater (three lake water, one pond water), one from freshwater sediment (lake sediment), and four from the terrestrial (two peat soil, two forest soil). The samples comprised 10,609,939 sequences amplified using the general eukaryotic primers 3NDf and 21R (Jamy et al., 2022). These primers targeted a segment of the 18S gene, the entire internal transcribed spacer (ITS), and a portion of the 28S gene. The amplicons were subsequently sequenced using PacBio’s Sequel II platform. No spike-in or positive control sequences were included in these samples. Before chimera filtration per-sample, all samples underwent primer trimming by cutadapt v4.4 (Martin, 2011), quality filtering with DADA2 v.1.32 (truncLen = 0, maxEE = 2, minQ = 3, rm.phix = FALSE), extraction of the full-length ITS region by ITSx (--nhmmer TRUE, -E 1e-2, --complement TRUE, --only_full TRUE), and dereplication by VSEARCH. Putative tag jumps were filtered out by UNCROSS2 (Edgar, 2018) within PipeCraft2 (Anslan et al., 2017) (options: f = 0.03, p = 1). We deployed 2,070,676 dereplicated ITS sequences for the chimera filtration processes (Fig. 1).

Figure 1 Workflow related to the empirical dataset.

ITS amplicon sequences initially were trimmed with cutadapt and quality filtered using the filterANDtrim function (maxEE = 2, minQ = 3) of DADA2. Subsequently, the full ITS region was extracted from the sequences using the ITSx software. Dereplication was performed using VSEARCH, followed by tag-jump filtering with UNCROSS2 (f = 0.03, p = 1). Chimera filtering was carried out using uchime_denovo, chimeras_denovo, and removeBimeraDenovo. The “Default” path refers to runs using the software’s standard parameters, while the “Adjusted” path uses the optimized parameters for uchime_denovo and chimeras_denovo that were identified from our simulated dataset analysis (see Table 1). The validation module based on BLASTn was used as a secondary step against the EUKARYOME database to identify and recover false positives from chimeric outputs and remove false negatives from nonchimeric outputs.

Detection of false-positive and false-negative chimeric reads

Reads filtered out during the chimera filtering step were subjected to BLASTn (Altschul et al., 1990) search (word size = 7; reward = 1; penalty = -1; gap opening cost = 1; gap extension cost = 2), using EUKARYOME v.1.9.2 as a reference database. When a query sequence had a high identity (≥99%) and high query coverage (≥99%) against the reference sequence, then the query read was considered a false-positive chimera. The remaining sequences in the chimeric set (after chimera filtering) were considered true chimeras. Reads that passed the chimera filtering process were also subjected to BLASTn search, where, in addition to EUKARYOME, the set of sequences from the corresponding sample (the sample from which the query originated) served as a reference database. To identify true chimeras, we first excluded self-hits, where the query and target sequence IDs were identical. For the remaining sequences, if a query had multiple alignments per best hit and the first high-scoring segment pair (HSP) had a query coverage below 85%, the sequence was classified as a true chimera. If such a chimera (sequence) was found within the non-chimeric output of the chimera filtering algorithms, it was flagged as a false negative chimera.

OTU-level comparisons between chimera filtering approaches

Output reads of each chimera filtration algorithm with default and adjusted settings (except removeBimeraDenovo) were clustered into operational taxonomic units (OTUs) using VSEARCH with a similarity threshold of 98%. Following this, chimeric and non-chimeric batches of sequences were subjected to BLASTn search against the EUKARYOME database to identify false-positive and false-negative chimeras. After recovering false positives and discarding false negatives from the chimera filtered dataset, the final OTUs set is referred to as “adjusted + FP-FN”. Clustering was done before chimera filtration to generate the “raw data OTUs”.

To account for differences in sequencing depth, we retained raw OTU count tables and converted OTU abundances to binary presence/absence (PA) data. Raw data OTUs were compared against those treated with various chimera removal methods, and default chimera filtering settings against the ones where false positives and false negatives were corrected. Furthermore, we performed all pairwise comparisons among chimera-filtering combinations (default vs. adjusted + FP-FN). For each table, sequence data were Hellinger-transformed using the decostand function from the vegan package v.2.6-8 (Oksanen et al., 2024) and Bray-Curtis dissimilarities computed using the vegdist function. After performing non-metric multidimensional scaling (NMDS), we used paired Procrustes tests to quantify discrepancies between ordinations. We assessed whether the impact of parameter tuning differed among chimera-filtering methods and sample types by fitting a linear mixed-effects model in lme4. The response variable was the Procrustes residual for each sample, and we treated both the filtering method (chimeras_denovo, removeBimeraDenovo, uchime_denovo) and the isolation source (forest soil, lake water, marine water, peat soil) as fixed factors. Sample identity was included as a random intercept to account for repeated measures across methods. We compared this full model to a reduced version that omitted the method × habitat interaction using AICc and a likelihood-ratio test (p < 0.001), confirming that the interaction term significantly improved model fit. Finally, we extracted marginal predictions and conducted pairwise contrasts among methods (holding habitat constant) to identify which algorithms showed the largest shifts in residuals when default parameters were replaced with adjusted + FP-FN settings. Visualizations were generated in ggplot2 v.3.5.1 (Wickham, 2016).

Kruskal–Wallis rank-sum test was utilized to assess whether different chimera removal methods had a significant effect on alpha diversity. Alpha diversity was assessed using Shannon’s index and the residuals from a linear model. Linear models were fitted with ‘chimera removal method’, ‘isolation source’ (e.g., soil, freshwater, marine), and their interaction as fixed effects. Residuals from these models were extracted and used as an adjusted alpha diversity measure. Statistical significance was defined at α = 0.05. All analyses were conducted in R v.4.4.2 (R Core Team, 2024) and RStudio v.2024.12.1.563 (R Core Team, 2024).

Characteristics of false-negative chimeras

To investigate why certain chimeric sequences escape detection (false negative chimeras), we examined their structural properties in the empirical dataset. We used Infernal v.1.1.5 (Nawrocki & Eddy, 2013) and the inferrnal R package v.0.99.8 (Furneaux, 2024) with the RF00002 covariance model from Rfam (https://rfam.org/family/RF00002) under default settings to identify the quantity and positions of the 5.8S rRNA gene within the ITS sequences. The 5.8S rRNA gene is a conserved component of the ITS region, and a genuine ITS sequence should contain exactly one 5.8S gene. However, if a sequence contains multiple 5.8S rRNA genes, it might be a chimera resulting from concatenating two or more ITS regions. We also examined the length distribution of the false negatives, as unusually long reads could further indicate concatenation artefacts.

Results

Chimera detection in the simulated dataset

The default settings of uchime_denovo detected 1,975 (79.5%) out of 2,484 simulated chimeric reads with no false-positive detections (F1 score = 0.87; Fig. 2, Fig. S1). The algorithm yielded the highest chimera detection performance (F1 = 0.89; 1,991 true chimeras and two false positives; Table S4) when the parameters were changed to abskew 3 (default is 2) and minh 0.09 (default is 0.28), while all other settings remained default (Table S4; Fig. 2, Fig. S1).

The chimeras_denovo algorithm with default settings detected 1,199 (48.3%) true simulated chimeras, but it also filtered out many false-positive chimeric sequences (11,088 reads; F1 score = 0.15). The best performance of the chimeras_denovo algorithm was achieved by setting the abskew value to 4 (default is 1), the chimeras_length_min value to 30 (default is 10), and the chimeras_diff_pct value to 0.9 (default is 0). This detected 1,447 true chimeric reads and 143 false positives (F1 score = 0.71; Table S5). The undetected chimeras (i.e., false negatives) from both algorithms’ (uchime_denovo and chimeras_denovo) outputs did not demonstrate any other special characteristics besides a few being abnormally long (>1,000) or short (<100) sequences (Table S3).

The removeBimeraDenovo algorithm was tested only with the default settings, yielding 1,130 (45.5%) true chimeric and 12,764 false positive reads (F1 score = 0.14). It failed to filter out true chimeric sequences where one of the parts was in reverse complementary orientation (compared to the parent sequence) and sequences from closely related species.

Figure 2 The F1 scores different algorithms for detecting chimeric sequences in a simulated dataset.

uchime_denovo, with adjusted settings (abskew 3, minh 0.09), achieved the highest F1 score (0.89). Other configurations, including removeBimeraDenovo and the default settings of chimeras_denovo, showed lower F1 scores of 0.14 and 0.15. This highlights the varying effectiveness of different chimera filtering methods.

Chimera detection in the empirical dataset

The uchime_denovo filtered out 12,114 reads (0.58%) as chimeric based on the default settings. Conversely, when the best setting (abskew 3, minh 0.09; as identified in the simulated dataset; adjusted settings) was applied, more reads were filtered out: 41,087 reads, comprising 2% of the dataset. The chimeras_denovo filtered out 498,437 reads (24.07%). Under adjusted settings (abskew 4, chimeras_length_min 30, chimeras_diff_pct 0.9), the proportion of discarded reads dropped notably to 99,673 sequences (3.85%). The removeBimeraDenovo algorithm with default settings filtered out 320,032 chimeric sequences (15.45%; Fig. 3). In total, only 151 reads were commonly flagged chimeric by all three methods with default settings; however, chimeras_denovo uniquely flagged more chimeras (270,207) than uchime_denovo and removeBimeraDenovo (Fig. 3). With adjusted settings for uchime_denovo and chimeras_denovo, 1,875 reads were commonly flagged as chimeras between these algorithms (Fig. 3).

Figure 3 Comparison of chimera filtering methods in the empirical dataset.

The UpSet plot illustrates the overlap among different chimera detection methods, highlighting the impact of parameter adjustments. Each vertical bar represents a unique group of sequences identified as chimeras by one or more methods; the dots connected by lines below each bar indicate which combination of methods detected these sequences as chimeras. The height of each vertical bar (y-axis) shows the number of chimeric sequences detected only by that specific combination of methods. The horizontal bar plot on the left side of the figure indicates the total number of chimeric sequences detected by each filtering method. Here, “default” refers to the software’s standard parameters. “Adjusted” refers to the optimized settings identified from the simulated dataset: abskew 3 and minh 0.09 for uchime_denovo, and abskew 4, chimeras_length_min 30, and chimeras_diff_pct 0.9 for chimeras_denovo. removeBimeraDenovo was run only with default settings. The results emphasize that certain parameter settings and methods detect unique chimeric sequences, underscoring the importance of method choice and parameter adjustment impact in identifying and removing chimeras from sequencing datasets.

Following chimera filtration, the chimeric sequences identified with default and adjusted settings were subjected to taxonomy annotation to identify false-positive chimeras. Using default settings, the false-positive rate in the putative chimeric reads set was 9.8% (1,052 reads) for uchime_denovo, 55.0% (276,346 reads) for chimeras_denovo, and 52.2% (167,102 reads) for removeBimeraDenovo (Fig. 4). Using adjusted settings, the false-positive rate increased to 19.9% (8,618 reads) for uchime_denovo, but it decreased to 38.6% (37,876 reads) for chimeras_denovo (Fig. 4). A comparative analysis of false positives across the three tools revealed that, using default settings, only 75 reads were shared among all methods (Fig. 5A). With adjusted settings (uchime_denovo & chimeras_denovo), the overlap increased to 90 reads, with removeBimeraDenovo (default settings) showing the largest count of unique false positives (Fig. 5C).

Figure 4 Doughnut charts of sequences classified as false-positive and false-negative by uchime_denovo, chimeras_denovo, and removeBimeraDenovo.

Default settings are the software’s standard parameters. “Adjusted settings” are the optimized configurations derived from the simulated dataset (abskew 3 , minh 0.09 for uchime_denovo; abskew 4, chimeras_length_min 30, chimeras_diff_pct 0.9 for chimeras_denovo). removeBimeraDenovo was only tested with default settings. Default settings compared with optimized configurations (adjusted settings) derived from the simulated dataset. The adjusted settings reduced false negatives while increasing false positives. The charts illustrate the trade-offs between sensitivity and specificity in each tool’s performance. Non-chimeras represent sequences with high identity (≥ 99%) and high coverage (≥ 99%) against the EUKARYOME database, while false-negative chimeras are chimeric sequences missed by the detection process.

Figure 5 Venn diagrams compare false positives (FPs; A, C) and false negatives (FNs; B, D) in empirical datasets identified by BLAST for outputs of uchime_denovo, chimeras_denovo (default and adjusted settings), and removeBimeraDenovo (default on “Default settings” refer to standard software parameters.

“Adjusted settings” refer to the optimized parameters for uchime_denovo (abskew 3, minh 0.09) and chimeras_denovo (abskew 4, chimeras_length_min 30, chimeras_diff_pct 0.9) derived from the simulated dataset analysis. With default settings, chimeras_denovo identified a high number of FPs (264,445; (A) and 1,275 FNs (B), while uchime_denovo identified 1,052 FPs (A) and 1,314 FNs (B). After parameter tuning, chimeras_denovo significantly improved in precision, reducing FPs substantially to 30,888 (C), while recall remained similar with 1,291 FNs (D). Conversely, adjusted settings for uchime_denovo displayed a sharp increase in FPs (rising from 1,052 to 8,618; (C) with only marginal improvement in FNs (decreasing slightly from 1,314 to 1,271; (D). removeBimeraDenovo, constrained to default settings, showed limited adaptability regarding the high number of FPs (156,130; A) and FNs (1,297; B) identified in these long-read datasets.

To determine the taxonomic composition of false-positive chimeras, we analyzed the phyla associated with each sample derived from the false-positive sequences identified by various chimera filtering methods. Using the uchime_denovo, sequences of Basidiomycota were most often misdetermined as chimeras across all habitats, with the highest relative rate in soil samples (Figs. S5A, S6A, Table S6). Ascomycota contributed strongly to false positives in freshwater and soil samples, while Glomeromycota were erroneously determined to be chimeric from soil samples (Figs. S5A, S6A, Table S6). In contrast, the chimeras_denovo method indicated that false-positive chimeras (Figs. S5B, S6B, Table S6) are mainly related to Dinoflagellata in freshwater samples, Arthropoda in soil samples, and Cercozoa in marine samples. The removeBimeraDenovo method misdetermined Basidiomycota and Ascomycota as chimeric in soil samples, but Annelida in marine samples. False-positive chimeras in freshwater samples were more evenly distributed among phyla (Figs. S5C, S6C).

The putatively nonchimeric outputs were evaluated for the presence of false-negative chimeras (chimeric sequences not identified during filtration; see ‘methods’). Under default settings, the rates of false-negative chimeras were consistent across various algorithms: 0.06% (1,314 from 2,058,562 reads) in uchime_denovo, 0.06% (1,275 from 1,572,239 reads) in chimeras_denovo, and 0.06% (1,297 from 1,750,644 reads) in removeBimeraDenovo (Fig. 4). With adjusted settings, these rates were slightly changed to 1,271 reads (total: 2,029,589, 0.06%) for uchime_denovo and 0.06% (1,291 from 1,971,003 reads) for chimeras_denovo. A comparison of shared false negatives showed that most sequences overlapped between methods (1,240 for default settings, and 1,238 for adjusted settings; Fig. 5). The empirical dataset’s false negative chimeric sequence length distribution exhibited a bimodal pattern with peaks at ca. 500 bp and 5,000 bp (Fig. S7). There was one group of sequences centered on the expected size of the ITS amplicon and another distinct group that consisted of extremely long reads, some longer than 3,000 bp. The results of Inferrnal for the 5.8S rRNA composition showed a relationship between sequence length and the number of 5.8S rRNA gene regions detected (Fig. S8). Most of these sequences were with two or more 5.8S hits. Specifically, 97.9% (1,218 of 1,244 reads) of false negatives from uchime_denovo and 95.5% from chimeras_denovo (1,218 of 1,275) with default settings. These figures remained high with adjusted settings, at 95.0% (1,207 of 1,271 reads) for uchime_denovo and 93.7% (1,210 of 1,291 reads). for chimeras_denovo. However, it was less pronounced in the removeBimeraDenovo, where 59.7% of false negatives (774 of 1,297 reads) had multiple 5.8s regions. These sequences had considerably higher median lengths (∼4,500 bp) than false negative sequences with no or one 5.8S hit (∼600 bp).

The impact of different chimera removal methods on OTU community structure was assessed using Procrustes analysis (Fig. S9). Procrustes correlations within each algorithm’s default versus adjusted + FP–FN (algorithm’s adjusted settings, as above, with recovering false positive and discarding false negative reads) pair remained very high (Procrustes R = 0.93–0.99, p = 0.001), with uchime_denovo demonstrating the strongest concordance (Procrustes R = 0.99; thus, lowest residuals, Fig. S10). With default settings, removeBimeraDenovo and chimeras_denovo filtered out high rates of false positives (52.2% and 55.0%, respectively), corresponding to the highest residuals (after Procrustes analysis) in the default versus adjusted + FP–FN comparison (Fig. S10). Similarly, all pairwise comparisons among default and adjusted + FP–FN chimera-filtering combinations yielded significantly high correlations (Procrustes R = 0.94 − 0.98, p = 0.001), but residuals were overall slightly lower between adjusted + FP–FN pairs (indicating reduced biases between algorithms; Fig. S10). The comparison of outputs from all chimera-filtering methods (both default and adjusted + FP–FN) with raw OTU tables (no chimera filtering performed) also showed significantly correlated sample positions in ordination space, but to a lesser extent than those observed among the chimera-filtered datasets (Procrustes R = 0.93–0.98, p = 0.001; Fig. S9A). The linear mixed-effects model was applied to the residuals of the default and adjusted + FP-FN comparison, including the interaction between the chimera filtering method and the isolation source (Fig. S11). Both the chimera removal method (F = 32.16, p < 0.001) and the isolation source (F = 11.54, p < 0.001) were significant factors. Importantly, the interaction between the method and the source was also highly significant (F = 11.90, p < 0.001). The chimeras_denovo had the largest residuals, particularly in forest soil (Estimate = 0.55) and lake water (Estimate = 0.58), whereas uchime_denovo was consistent, and residuals remained close to zero. Moreover, average pairwise comparisons confirmed that uchime_denovo had significantly lower residuals than both chimeras_denovo (Estimate = −0.24, p < 0.001) and removeBimeraDenovo (Estimate = −0.15, p < 0.001).

Kruskal–Wallis rank-sum tests revealed no significant effect of the chimera removal method on overall OTU richness or the Shannon diversity index (Figs. S12, S13). Across all three chimera filtering algorithms, correcting false positives mainly affected the sequence abundance of dominant OTUs. Nevertheless, this process recovered hundreds of unique OTUs, but most could only be assigned to higher taxonomic levels (e.g., kingdom or phylum). Specifically, for chimeras_denovo adjusted + FP-FN, 158 OTUs were gained, yielding three additional genera (from kingdoms Rhizaria, Haptista, and Metazoa; the remaining 155 OTUs classified to the kingdom/phylum level; Table S7). For removeBimeraDenovo + FP-FN, only 13 new OTUs were gained, adding a new metazoan genus. The largest OTU gain arose within the uchime_denovo adjusted + FP-FN dataset, which hosted 346 unique OTUs, and added four new fungal species and one genus (Table S7).

Discussion

This study assessed the performance of three chimera filtering algorithms, uchime_denovo, chimeras_denovo, and removeBimeraDenovo. We compared their chimera detection accuracy rates by applying these tools to simulated and empirical ITS amplicon datasets. uchime_denovo (with adjusted settings) demonstrated the highest accuracy on simulated data. However, these adjusted settings based on simulated data did not directly translate into higher efficiency in filtering chimeras from empirical data. This highlights the challenges in chimera removal from complex datasets and calls for secondary validation strategies to improve precision.

Performance insights from parameter optimization

The default settings of uchime_denovo demonstrated relatively high precision in the simulated dataset, where adjustments resulted in a modest gain (Fig. 2). On the other hand, the efficacy of chimeras_denovo was considerably influenced by calibration. Its default settings generated a remarkably elevated incidence of false positives in the simulated dataset, which resulted in low precision performance (Fig. 2). Thus, the parameter adjustments of chimeras_denovo notably increased the performance. Here, we applied removeBimeraDenovo only with default settings, and its performance was similar to the outcome of the chimeras_denovo with default settings (Figs. 2, 4). This indicates that the default parameters of chimeras_denovo and removeBimeraDenovo may be too aggressive by incorrectly classifying natural sequence variation as chimeric ones.

A recent benchmarking study has shown that filtering PacBio HiFi reads with adjusted parameters of uchime3_denovo VSEARCH achieves high accuracy in resolving ASVs with a minimal number of false positive chimera detections (Overgaard et al., 2024). Similarly, our findings suggest parameter adjustments can improve precision while maintaining recall. However, our study also highlights that parameter optimization is not always straightforward. Switching from default to adjusted settings frequently improved performance by reducing false negatives, but mostly at the expense of increasing the rates of false positives. We also found that performance depends on the specific dataset, as optimal adjustments on our simulated dataset did not directly translate to the same efficiency on the empirical data. Similar context-dependent variability has been reported previously, where Edgar et al. (2011) demonstrated that UCHIME’s filtering efficiency can decline when applied to highly complex environmental samples, particularly where sequence similarity among parental taxa and chimera breakpoints complicates detection. Additionally, in the empirical dataset, the differences between corresponding sample points as measured in an OTU community ordination space (Procrustes residuals) revealed the significant effect of the chimera filtering method and isolation source. The method-specific nature of chimera detection was also demonstrated on the limited overlap of identified chimeric sequences among algorithms (Fig. 3).

Secondary validation

Annotating the discarded sequences by chimera filtering algorithms revealed that a substantial proportion of those were falsely labeled as chimeras (Fig. 4). The same was found from ITS2-based mock community analyses, where high rates of falsely identified chimeras were identified using UCHIME (Aas, Davey & Kauserud, 2017). In our study, surprisingly, an algorithm originally optimized for long-read data, chimeras_denovo, detected more false-positive chimeras than uchime_denovo. Similarly, the removeBimeraDenovo yielded high rates of false-positive chimeras, which could be ‘rescued’. However, the correction for false positives across all three algorithms resulted only in a slight gain of new OTUs that would have been discarded without secondary validation of false positives. Correspondingly, the alpha diversity metrics were not significantly different between chimera filtered datasets that either incorporated the rescue of false positives or not (Fig. S12). Similarly, the Procrustes analyses between all pairs of chimera-filtered datasets demonstrated high and significant correlations (Procrustes R = 0.94−0.98, p = 0.001), indicating no significant impact of false-positive chimeras’ exclusion on broad-scale community analyses. Likewise, analyzing the performance of UCHIME upon fungal ITS2 sequences, Aas, Davey & Kauserud (2017) concluded satisfactory results with the default chimera filtering approach.

The proportion of identified false-negative chimeras in the filtered datasets was notably smaller compared with false positives. Many false negatives were unusually long and, in several cases, contained multiple hits to the 5.8S rRNA region. This suggests that some chimeras in our dataset were not generated during PCR amplification but likely emerged during PacBio library preparation. The formation of long chimeric constructs during SMRTbell adaptor ligation can evade standard chimera detection algorithms (Fichot & Norman, 2013; Griffith et al., 2018). Although sequences outside the expected amplicon length can be filtered out during, for example, the quality filtering step, our analysis did not apply a maximum length threshold (because of the highly variable length of ITS region across eukaryotes; Yang et al., 2018). However, the relative abundance of these artifacts was low, suggesting that they hinder the community-level analyses even less than false positives. Nevertheless, reporting those as valid OTUs may create undesirable biases in precise richness analyses and generate erroneous metabarcoding feature entries when those are deposited, for example, in PlutoF (or similar) biodiversity data management platforms (Abarenkov et al., 2010).

The utility of secondary validation aligns with findings from studies that emphasize the value of hybrid approaches in improving long-read data analyses (Christel et al., 2023; Lu et al., 2023; Rué et al., 2023). Therefore, few metabarcoding pipelines, such as PipeCraft2 and BIOCOM-PIPE (Djemiel et al., 2020), have implemented false-positive chimera rescue methods based on the secondary validation against a reference database. However, the reliance on reference-based validation is hampered for taxa without reference sequences. Another strategy implemented in the Natrix (Welzel et al., 2020) and NextITS pipeline (Mikryukov, Anslan & Tedersoo, 2025), for false-positive chimeras, involve retrieving sequences initially considered as chimeras but are present in multiple samples because the formation of identical chimeras in multiple independent PCR reactions is highly unlikely. This strategy, however, requires careful filtering of tag-switching errors to avoid rescuing true chimeras that happen to leak to multiple samples. With the rising awareness that error-free chimera filtering is not achievable (Edgar, 2016), we can expect various secondary validation strategies to be incorporated into metabarcoding pipelines in the future.

In addition to secondary validation strategies and/or adjusting chimera filtering algorithm parameters, using upstream solutions in library preparation can be beneficial for the chimera burden. For instance, implementing Unique Molecular Identifiers (UMIs) in the amplicon sequencing process can improve chimera removal by filtering out non-matching UMI amplicons, regardless of the long-read sequencing technology used (Karst et al., 2021). Furthermore, the use of dual UMIs eliminates chimeras during the data processing step, making the de novo filtering discussed here less critical for both PacBio and Oxford Nanopore Technologies (ONT) when per-UMI coverage is adequate (Overgaard et al., 2024).

Implications of fine-tuning the chimera filtering process

We found that the chimera filtered vs. non-filtered datasets demonstrated significant and high correlations in an ordination space (Procrustes analyses on NMDS axes). Therefore, although the broad-scale community structure in an ordination space remains largely preserved, chimera filtering is advisable (Tedersoo et al., 2022). Further fine-tuning of the algorithm’s parameters, including correcting for false positives and false negatives, however, seems to have little impact on the final overall data structure. Importantly, the degree of correspondence between OTU datasets processed with default versus adjusted (+FP-FN) chimera filtering settings varied depending on the DNA isolation source (i.e., sampling substrate; Fig. S9). As noted above, chimera filtering may be more challenging in high-complexity environmental samples; therefore, suggesting higher importance of secondary validation in, for example, forest soil datasets compared with marine water datasets (Figs. S9, S11). While our study focused on PacBio HiFi reads, the difficulties of de novo chimera detection may likely extend to reads generated with other sequencing technologies with greater challenges expected in datasets with more complex communities. Taken together, these results indicate that while refining the chimera filtering process does not radically alter the overall structure of the data, secondary validation steps help to balance the methodological biases.

Conclusions

While this study was based on analyses of the ITS region, the observed chimera filtering algorithm performances may not be directly linked to other marker genes, because chimera formation and its detection efficiency can be influenced by marker-specific properties. Our analyses on the simulated dataset highlighted notable performance differences among chimera filtering algorithms, suggesting that the default settings of an algorithm are not equally appropriate for all datasets. Although the OTU-level analyses in the empirical dataset revealed that overall richness and community patterns remained highly conserved regardless of the filtering strategy, implementing secondary validation approaches in parallel could maximize performance by minimizing individual algorithmic biases. Advanced approaches, such as deep learning algorithms, could enhance chimera detection by incorporating both sequence context and abundance patterns to address the current limitations of existing tools. Nevertheless, while the loss of false positives may be acceptable in broad-scale ecological analyses, excluding easily identifiable false negatives, such as those with unusually long sequences, would be appropriate.

Supplemental Information

Supplemental Information 1 Supplementary Materials

Supplemental Information 2 Taxonomic levels of the sequences used for simulation by the SimLord simulator

Cutadapt v.4.4 (Martin, 2011) was used to trim the full ITS region from them. The primers used were ITS1degenerate (KCN GTW GGW GAA CCW GC) and ITS4ngsUni+ (CAT ATH ANT AAG SSS AGG), allowing 2 mismatches, and primer overlap was set to the maximum length of the primer minus two base pairs.

Supplemental Information 3 The scripts related to the simulation of data and simulated chimeric reads in the GitHub repository

Supplemental Information 4 Simulated chimeric sequences details

The details of the generation of 2,484 simulated chimeric ITS reads designed to benchmark chimera detection algorithms. Each entry specifies a unique chimera identifier, the two parental sequence IDs drawn from the EUKARYOME v1.9.2 database, and the corresponding breakpoint positions determined based on ITS-region-specific probabilities. The table further lists the parental sequence lengths, the resulting chimera length, and the abundance constraints that ensure the artificial chimeras comprise approximately 6.2% of the total dataset, thereby maintaining realistic sequence properties. These parameters, derived from a custom Python simulation script paired with SimLoRD-generated variability, ensure that the simulated artifacts mimic PCR-induced anomalies typical in long-read amplicon sequencing.

Supplemental Information 5 Chimera detection of uchime_denovo under different settings for the simulated dataset

The dataset comprises 2,484 true chimeras; the total number of reads in the dataset was 46,954. The settings were adjusted to find this algorithm’s best combination (highest F1 score). The F1 score is a statistical measure that evaluates the performance of an algorithm by considering both specificity and sensitivity. Initially, each setting’s value was changed, and later, the combination of various settings shifted to achieve better results (–abskew 3 –minh 0.09), which is more true chimera detection with the minimum false positives in our search for the optimal combination. False positives are non-chimeric sequences misidentified as chimeric sequences by the algorithm. In this algorithm, we achieved the best results when settings were set to –abskew 3 –minh 0.09 and other settings as default.

Supplemental Information 6 Chimera detection ratios for the chimeras_denovo under different settings for the simulated dataset

The dataset comprises 2,484 true chimeric sequences; the total number of reads in the dataset was 46,954. The settings were adjusted to find this algorithm’s best combination (highest F1 score). The F1 score is a statistical measure that evaluates the performance of an algorithm by considering both specificity and sensitivity. Initially, each setting’s value was changed, and later, the combination of various settings shifted to achieve better results in our search for the optimal combination. False positives are non-chimeric sequences misidentified as chimeric sequences by the algorithm. This algorithm achieved the best results when settings were set to –abskew 4 –chimeras_diff_pct 0.9 –chimeras_length_min 30 and other settings as default.

Supplemental Information 7 The total abundance of phylum-level sequences in chimera filtering that had a good match against the EUKARYOME database

The total abundance of phylum-level sequences in chimera filtering that had a good match (>=99%) against the EUKARYOME database after the chimera filtering step with uchime_denovo, chimeras_denovo, and removeBimeraDenovo. Other (Eukaryotes) and other (Fungi) are related to the phyla because they have very low abundance, so they are categorized in these groups.

Supplemental Information 8 Taxonomic classifications (kingdom to species) are presented for genera and species uniquely identified in the adjusted + FP-FN OTU datasets compared to the default outputs of each chimera-filtering method

Only classified taxa are included; empty species fields denote the absence of species-level identification.

The authors acknowledge CSC–IT Center for Science, Finland, and High Performance Computing Center of the University of Tartu (UTHPC) for computational resources.

Additional Information and Declarations

Competing Interests

Author Contributions

Data Availability

The authors declare there are no competing interests.

Ali Hakimzadeh conceived and designed the experiments, performed the experiments, analyzed the data, prepared figures and/or tables, authored or reviewed drafts of the article, and approved the final draft.

Vladimir Mikryukov analyzed the data, prepared figures and/or tables, authored or reviewed drafts of the article, and approved the final draft.

Martin Metsoja analyzed the data, prepared figures and/or tables, and approved the final draft.

Leho Tedersoo conceived and designed the experiments, authored or reviewed drafts of the article, and approved the final draft.

Sten Anslan conceived and designed the experiments, authored or reviewed drafts of the article, and approved the final draft.

The following information was supplied regarding data availability:

The data is available in the Supplemental Files.

The data is available at GitHub and Zenodo:

- https://github.com/alihkz94/long-chimeric-reads-project.

- Ali Hakimzadeh, & Vladimir Mikryukov. (2025). alihkz94/long-chimeric-reads-project: Long chimeric reads analysis scripts (v1.0.0). Zenodo. https://doi.org/10.5281/zenodo.17512355.

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
