# Peer review of "Are we throwing away good data? Evaluation of chimera detection algorithms on long-read amplicons reveals high false-positive rates across algorithms"

_PeerJ, doi:10.7717/peerj.20456_

## Round 0.1 · original submission · Major Revisions

· Academic Editor

Major Revisions

Title: Are we throwing away good data? Evaluation of chimera detection algorithms on long-read amplicons reveals high false-positive rates across algorithms
Authors: Hakimzadeh et al.

Dear Dr. Anslan,

The reviewers acknowledge the relevance and timeliness of this work, particularly given the growing use of long-read technologies (PacBio, ONT) in biodiversity and metabarcoding research. The benchmarking approach is carefully designed, using both simulated and empirical ITS datasets, and highlights important issues of false positives and negatives in chimera detection. However, both reviewers raise substantive concerns regarding scope, generalizability, robustness, and methodological choices. These concerns must be addressed before the manuscript can be considered for publication.

Major Points Requiring Revision

Marker generality and applicability

The study is limited to the ITS region. Please discuss explicitly how findings may (or may not) extend to other barcode markers (16S, 18S, COI). Differences in sequence composition and chimera formation rates could significantly affect tool performance.

Parameter robustness

uchime_denovo performed best with adjusted parameters derived from simulated datasets. Please assess or at least discuss how robust these parameters are across diverse empirical contexts, taxonomic compositions, and sequencing depths.

Library preparation artifacts

The identification of multi-5.8S artifacts is an important finding. However, reviewers request stronger validation. If spike-ins or positive controls were not performed, please acknowledge this explicitly and discuss the limitations.

Generality of tool selection

The benchmark focuses only on three algorithms. A justification is needed for excluding other widely used chimera detection approaches. At minimum, expand the rationale for focusing on these three tools and clarify whether the results should be seen as tool-specific.

Choice of simulated datasets

Reviewers question why newly generated simulated datasets were used instead of, or in addition to, publicly available ones (e.g., from the original UCHIME paper). Please justify this choice and explain how your design ensures fairness and generalizability.

Validation strategies and alternatives

The BLAST-based secondary validation is computationally costly. Reviewers suggest discussing or testing alternatives such as machine learning classifiers, heuristics (e.g., length filters), or rescue strategies based on multi-sample occurrence.

ONT applicability

Since Oxford Nanopore remains widely used and differs significantly from PacBio in error profiles (e.g., homopolymers), please include at least a discussion on whether the conclusions are expected to hold for ONT data. If possible, consider preliminary ONT benchmarking.

Dataset and tool coverage

For a comprehensive benchmark, reviewers expect testing across more datasets and tools. If expansion is not feasible, justify clearly the scope and limitations of the study.

Minor Points

Figures and workflow clarity

Improve the pedagogical clarity of Figure 2 (workflow) by including more technical details and enhancing the figure legend.

Clearly explain “Default Settings” vs. “Adjusted Settings” in all figure legends.

Language and style

Several minor phrasing issues were noted. For example:

Line 51: change “in delimiting between species” → “in delineating species.”

Line 437: change “not uniformly most appropriate” → “not equally appropriate.”

A thorough language edit by a fluent English speaker is strongly recommended.

Both reviewers converge on the need for expanded scope, stronger validation, and clearer justification of methodological choices. The paper addresses an important question in long-read metabarcoding, but significant revisions are required to strengthen its rigor and generalizability. Please revise the manuscript accordingly, providing a detailed point-by-point response to all reviewer comments.

**Language Note:** The Academic Editor has identified that the English language must be improved. PeerJ can provide language editing services - please contact us at [email protected] for pricing (be sure to provide your manuscript number and title). Alternatively, you should make your own arrangements to improve the language quality and provide details in your response letter. – PeerJ Staff

·

Basic reporting

This manuscript presents a thorough benchmarking study comparing three chimera detection algorithms for long-read metabarcoding data. The work is timely given the increasing adoption of third-generation sequencing in biodiversity studies, and the methodology is rigorous with both simulated and empirical datasets. The findings convincingly demonstrate that uchime_denovo outperforms other methods, particularly when parameters are optimized. While there are still some major concerns about the method design.

Experimental design

Major comments:
1. The study focuses exclusively on ITS region analysis - how generalizable are these findings to other commonly used barcode markers (e.g., 16S, 18S, COI), given known differences in sequence characteristics and chimera formation rates?

2. While uchime_denovo showed superior performance, its adjusted parameters (abskew 3, minh 0.09) were optimized on simulated data. How robust are these settings across different empirical datasets with varying taxonomic compositions and sequencing depths?

3. The manuscript identifies library preparation artifacts (multi-5.8S sequences) as a major source of false negatives. Were any positive controls or spike-in experiments performed to validate these findings and quantify their prevalence in PacBio workflows?

Validity of the findings

4. Given the high computational costs of secondary validation (BLAST searches), did the authors explore any machine learning approaches or heuristic filters (e.g., length thresholds) that could efficiently identify problematic sequences without reference databases?

5. The study focuses on PacBio sequencing, which produces highly accurate HiFi reads. Given that ONT sequencing typically has higher error rates (especially in homopolymer regions), how might these chimera detection algorithms perform differently on ONT data? Have the authors tested or considered benchmarking these tools on ONT datasets, and if not, would they expect similar performance trends?

Additional comments

Minor comments:
1. The current workflow diagram (Figure 2) provides a linear overview of the pipeline but could be enhanced to improve the pedagogical utility of each section. The authors are suggested to list more technical details for their workflow in the figure description.

2. The authors should provide clearer explanations of the "Default Settings" and "Adjusted Settings" in all figure descriptions to enhance readers' understanding.

Reviewer 2 ·

Basic reporting

Hakimzadeh et al present the results of a benchmark study of 3 de novo chimera detection algorithms in the context of long-reads (Pacbio reads) and to assess the predictive quality of these algorithms' detections. This study does show how challenging it is to accurately detect a chimera in complex datasets.

Many issues were found in the current version of the manuscript, listed below:
1) Major issues:
- In Results:
Why benchmark only 3 Methods/Tools while there are many more available? It's important to justify the reason why and make sure the presented findings are not tool-specific.

- In Materials and Methods:
Line 87: The Authors created their own simulated datasets. Why did they not use publicly available simulated datasets?
How about the simulated datasets published in the UCHIME paper or other relevant papers?
No explanation justifying this decision was provided. It’s important to use independent simulated datasets to ensure the benchmark is unbiased and fair. Moreover, the sets of simulated datasets should be large/diverse enough to cover datasets that are commonly/frequently published in life science studies.

2) Minor issues, especially about typos/phrasing, were found. I recommend you include more reviews from a native English speaker. Indeed, here are a few examples :
- Introduction:
Line 51: “ in delimiting between species” -> “in delineating species “

- Conclusions
Line 437: “suggesting that the default settings of an algorithm are not uniformly most
appropriate for all datasets” -> “suggesting that the default settings of an algorithm are not equally
appropriate for all datasets.”

Experimental design

More Tools and datasets should be included for a comprehensive benchmark and results interpretation.

Validity of the findings

Findings should rely on a sufficiently high number of tools and datasets.

---

## Round 0.2 · accepted · Accept

· Academic Editor

Accept

Congratulations on the acceptance of your manuscript.

·

Basic reporting

The authors have revised my comments. I appreciate the authors' efforts in responding to the critiques so effectively. The manuscript is now much improved and, in my view, meets the standards for publication. I recommend its acceptance in its present form.

Experimental design

The experimental design is methodologically sound

Validity of the findings

The validity of the findings is strongly supported by the experiment